

# In or out: Response slowing across housing conditions as a measure of affect in three Western lowland gorillas (*Gorilla gorilla gorilla*)

Molly McGuire[1,2] and Jennifer M. Vonk[2]

[1] Zoo Miami, Miami, FL, USA
[2] Department of Psychology, Oakland University, Rochester, MI, USA

## ABSTRACT

**Background:** Individuals experiencing negative affect have shown response slowing, a longer latency to respond in relation to baseline, when presented with aversive stimuli. We assessed response slowing in three male gorillas housed in a bachelor group as a function of daytime and nighttime housing arrangements.

**Methods:** In both experiments, three gorillas were rewarded for touching a single image (baseline, non-threatening gorilla or threatening gorilla) on a touchscreen. In Experiment One, they completed 48 50-trial sessions across combinations of three nested daytime and three nighttime conditions. In Experiment Two, they completed eight 50-trial sessions with novel stimuli across two daytime conditions, which were nested within two nighttime conditions. Housing conditions represented different amounts of space and degree of choice. We predicted that the gorillas would show response slowing to threatening stimuli when space and choice were restricted.

**Results:** We did not observe response slowing in Experiment One, although daytime and nighttime conditions interacted to predict response latencies. The gorillas responded more slowly when they had access to indoors and outdoors overnight compared to when they were in their stalls or together in an indoor habitat, but only if they had been given access to both indoors and outdoors or locked in the indoor habitat the day before. In Experiment Two, the gorillas did show response slowing to threatening stimuli, but this pattern did not interact with housing conditions. Our results, although limited by a small sample, are somewhat consistent with those of a previous study that did not find significant response slowing for apes as a function of aversive testing conditions, although the procedure has been effective in identifying dysregulated fear (high fear in low threat conditions) in macaques. The utility of this paradigm for testing affect in apes awaits further evaluation.

# INTRODUCTION

A recent trend in animal welfare is the assessment of affective states to complement the traditional emphasis on behavior and physiology. Affective states may be the most elusive aspect of welfare to evaluate given that they are not directly observable and any

Corresponding author
Jennifer M. Vonk,
vonk@oakland.edu

behavioral measures are subject to interpretation. Researchers have increasingly made use of cognitive bias tests, such as judgement bias tests (*Harding, Paul & Mendl, 2004*; *Mendl et al., 2009*; *Paul, Harding & Mendl, 2005*), to assess when animals are optimistic or pessimistic, but these tests often reveal contradictory patterns of results and may not assess affective states at all (*Perdue, 2017*). Furthermore, they often require long periods of training and animals may fail to reach training criteria, making it impossible to conduct the tests that assess affect (*McGuire & Vonk, 2018*; *McGuire et al., 2017*; *McGuire, Vonk & Johnson-Ulrich, 2017*). Despite some question about their interpretation, previous studies have shown that judgement bias tests can be sensitive to affect changes in domestic animals following manipulations of housing conditions (for review see *Baciadonna & McElligott, 2015*). We were interested in the affect of three gorillas in a bachelor group as a function of habitat available to them throughout the day and night. Given the challenges of other paradigms used to assess affective state, we adopted a response slowing procedure (*Bethell et al., 2016*, *2019*) both to assess its validity for assessing affect in apes, and for potentially evaluating the effects of housing conditions on the affect of these three gorillas.

Bethell et al.'s (2016, 2019) novel response slowing task assesses differences in reaction time to three categories of stimuli (baseline, non-threatening, threatening/aversive), with the idea that responses to threatening stimuli should be slower than to baseline and non-threatening stimuli, particularly under conditions of threat. Individuals showing this pattern of response are deemed sensitive to threat, which is assumed to be associated with negative affect. The first phase of this method involves a minimal training phase. In *Bethell et al.'s (2016)* study, macaques were rewarded for touching a grey square presented on a touchscreen in three spatial locations (left, right and center). In the testing phase, researchers continued to present the grey squares (to determine a baseline of responding), but also introduced grey squares containing pictures of faces of unfamiliar macaques that were either threatening (direct eye contact) or non-threatening (averted gaze). Macaques that had been subjected to recent veterinary exams, which are considered stressful, displayed a slower latency to touch threatening stimuli compared to baseline. The test was later validated by showing that animals that showed the strongest response to the human intruder task (HIT)—a more invasive assessment of behavioral freezing in response to an unfamiliar human—also showed the greatest behavioral inhibition to the touchscreen task, in that they were most likely to not touch the threatening stimuli (*Bethell et al., 2019*). However, responses in the HIT did not predict response slowing to the aversive stimuli in the computer task.

Although *Bethell et al. (2019)* did not obtain strong validation for response slowing as a measure of affect, the task is worth investigating further given that it can be implemented quickly without the extended period of training that characterizes most of the cognitive bias assessments available to date. This procedure has also been used with gorillas, chimpanzees and Japanese macaques (*Cronin et al., 2018*). These primates were tested during an air show over the zoo where they were housed, as well as under quieter conditions. The macaques showed the predicted response slowing to threatening faces

relative to baseline in the noise condition compared to the control conditions, but this pattern was not well evidenced with the apes.

Other researchers have successfully shown that the affect of apes can be captured in a different paradigm using response slowing as a measure. In an emotional Stroop task, human participants are asked to read various words and identify the font color that the words are presented in. The subject of these words may be negative or neutral/positive. People that have been diagnosed with clinical anxiety or depression take longer to identify the color of negative words relative to control subjects (*Williams & Nulty, 1986*). Additionally, researchers were able to successfully induce this response slowing by exposing high-anxiety subjects to a mood-manipulation expected to produce a negative mood state (*Richards et al., 1992*). Adapting this emotional Stroop procedure to chimpanzees, *Allritz, Call & Borkenau (2016)* trained chimpanzees to respond to images placed within differently colored borders with one color indicating reward. When negative images, such as images of veterinarians, were placed within the previously rewarded borders, chimpanzees responded more slowly, especially if they had recently experienced an aversive veterinary procedure. This study provided a proof of concept indicating that response times can be used to indicate negative affect in apes. However, *Bethell et al. (2019)* showed that macaques responded more slowly to pictures of objects than human intruders. This unexpected result, coupled with the fact that the response slowing procedure has not yet been shown to be effective with apes, suggests that the response slowing paradigm requires further validation for use with apes.

Previous studies have indicated the effects of different housing conditions on behavior in monkeys (*Fontenot, Wilkes & Lynch, 2006*) and apes (*Hoff et al., 1997*; *Lukas, Hoff & Maple, 2003*). Of relevance to the current study, *Aureli & De Waal (1997)* examined the behavior of five groups of captive chimpanzees as a function of whether they had access to indoor and outdoor areas of their habitat or were locked into indoor areas, cutting available space in half. They used behavioral measures as indicators of anxiety and showed that these behaviors were elevated under the higher-density indoor only condition. However, in a later paper, *De Waal (1989)* introduced the idea of behavioral adjustment to discuss how apes mitigated the presumed negative effects of increased density. Rhesus macaques living in long-term stable groups in captivity showed similar rates of aggression to those living at a field station (*Aureli & De Waal, 1997*). Others have also found that primates use strategies to reduce stress and aggression when conditions become more crowded (*De Waal, Aureli & Judge, 2000*; *Duncan et al., 2013*; *Nieuwenhuijsen & De Waal, 1982*). These studies may show that animals mitigate against increased levels of outward aggression through the use of behavioral tactics, but this does not mean that they do not experience higher levels of stress. In fact, *De Waal, Aureli & Judge (2000)* suggest that efforts to avoid escalations in aggression may come at the cost of increased stress and anxiety. Therefore, it is important to attempt to assess internal affective states in addition to outward behavioral outcomes as a response to variable housing conditions. Like *Cronin et al. (2018)*, we were interested in the effects of environmental changes on a group of captive gorillas, but we assessed effects of housing arrangements.

We studied a bachelor group of three relatively young silverback gorillas that experience intermittent aggression due to a shifting dominance hierarchy. These gorillas have access to two different indoor and outdoor habitats at different times of the year. In addition, depending on husbandry and management requirements, the gorillas either spent the night together in the dayroom (the large 205 m$^2$ indoor habitat) or they spent the night individually in a smaller space (13 m$^2$). At times, they had access to both indoor and outdoor areas during the day and at night, resulting in nine different daytime/nighttime housing conditions over the same year long period. We predicted that, during the warmer months, when the gorillas had access to the outdoor habitat, they would display less response slowing due to the positive effects of additional space (i.e., less intense competition for prime space and resources) and increased environmental complexity compared to when they were unable to access this portion of the habitat. Additionally, the gorillas' sleeping arrangements may impact their affect. As adult silverback males, these individuals spend a relatively high proportion of time vying for dominance (J. M. Vonk, 2012–2017, personal observation) and it may be that they experience a tradeoff concerning these two sleep locations (when there is less available space, it is also easier to monitor the location of conspecifics, or, when housed in stalls overnight, the gorillas were isolated and not vulnerable to threat displays from conspecifics). Gorillas may feel more comfortable in one location over the other, which may impact their affect in testing the following day. In addition, given the benefits of choice demonstrated by previous studies of great apes (*Kurtycz, Wagner & Ross, 2014*) and bears (*Owen et al., 2005*; *Ross, 2006*), we expected less response slowing to threatening stimuli when the gorillas had access to both indoor and outdoor spaces (accessed). Gorillas, like macaques, interpret direct eye contact as a threat signal (*Schaller, 1963*), and thus, might be expected to respond similarly in the task using direct gaze as a threatening cue. Thus, we assessed response slowing by presenting baseline, threatening, and non-threatening stimuli to three male gorillas across these housing conditions in two separate experiments.

# EXPERIMENT ONE

## Methods

### Subjects

Subjects were three adult male western lowland gorillas (Chipua/ "Chip", Pendeke/ "Pende" and Kongo) between the ages of 17–19 years old. The gorillas were housed together at the Detroit Zoo, in Royal Oak, MI. It was believed that Kongo was the dominant male at the time of testing, although Chip had been dominant in the past (keeper observations). Data collection took place in animal holding areas that were inaccessible to the public, three mornings each week at 07:00 h over a period of 2 months. Gorillas were separated into individual holding areas prior to data collection. These gorillas had previously been trained to use the touch screen for other studies (*McGuire & Vonk, 2018*; *McGuire et al., 2017*; *McGuire, Vonk & Johnson-Ulrich, 2017*; *Vonk et al., 2014*). Training and testing with these animals was approved by the IACUC of Oakland University

(12063-R1-A1), and the activities were presented as a form of cognitive enrichment in addition to serving to assess affect.

### Materials

The experimental apparatus consisted of a durable Panasonic Toughbook CF19 Laptop and 19″ VarTech Armorall capacitive touch-screen monitor welded inside a rolling LCD panel cart. Using a 1.2 m by 1.2 m plywood ramp, the apparatus was positioned flush against the mesh of the gorilla's holding stalls. The gorillas were given small pieces of chow, various fruits and vegetables from their breakfast trays as a reward for correct responses (in this case, simply touching the stimulus). This task was programmed using Inquisit Version 3 for Windows. All stimuli were presented on a black background. Baseline stimuli were blank grey squares of dimensions 400 by 300 MP. Test sessions included both threatening (3) and non-threatening (3) stimuli in addition to the baseline stimuli.

To create the three threatening stimuli, photographs of unfamiliar male Western lowland gorillas posturing in an aggressive manner (direct gaze, rigid posture etc.) were placed within the familiar grey baseline square (see Fig. 1). To create three non-threatening stimuli, photographs of unfamiliar males engaged in nonaggressive behavior (eyes averted, relaxed posture etc.) were placed within the familiar grey baseline square (see Fig. 1).

### Procedure

#### Training

The training phase consisted of ten 15-trial sessions. On each trial, the baseline stimulus was presented at one of three locations (left, center, or right). The stimulus appeared five times at each location in random order. When the gorillas touched this stimulus, a melodic tone was played and they received a single piece of chow or small piece of produce in a PVC feeder affixed to the side of the LCD cart. The next stimulus appeared after a 750 MS ITI immediately after a stimulus was selected.

#### Testing

Each gorilla was presented with 48 testing sessions across all possible housing arrangements. Testing took place between October 2016 and October 2017. The conditions were determined based on where the gorilla had spent the previous day (locked *indoors*, which consisted of being confined to either the north day room (NDR) or the south day room (SDR), locked outdoors in either the north or south outdoor habitat, or accessed to both indoors and outdoors) and where they had spent the night (locked indoors overnight in either the NDR or SDR/dayroom or locked in individual *stalls* or given *access* to the dayroom and outdoors). We treated indoor and outdoor conditions as separate factors because the issue of space may be qualitatively different when sleeping during the night and when foraging during the day.

Both outdoor habitats (seen in Fig. 2) included grassy substrates, living, dead and artificial trees and a termite mount. The larger habitat also included a shallow pool. Daytime conditions were nested within nighttime conditions so that we attempted to test each type of daytime condition equally often following each type of nighttime condition, but the conditions were unbalanced based on important husbandry

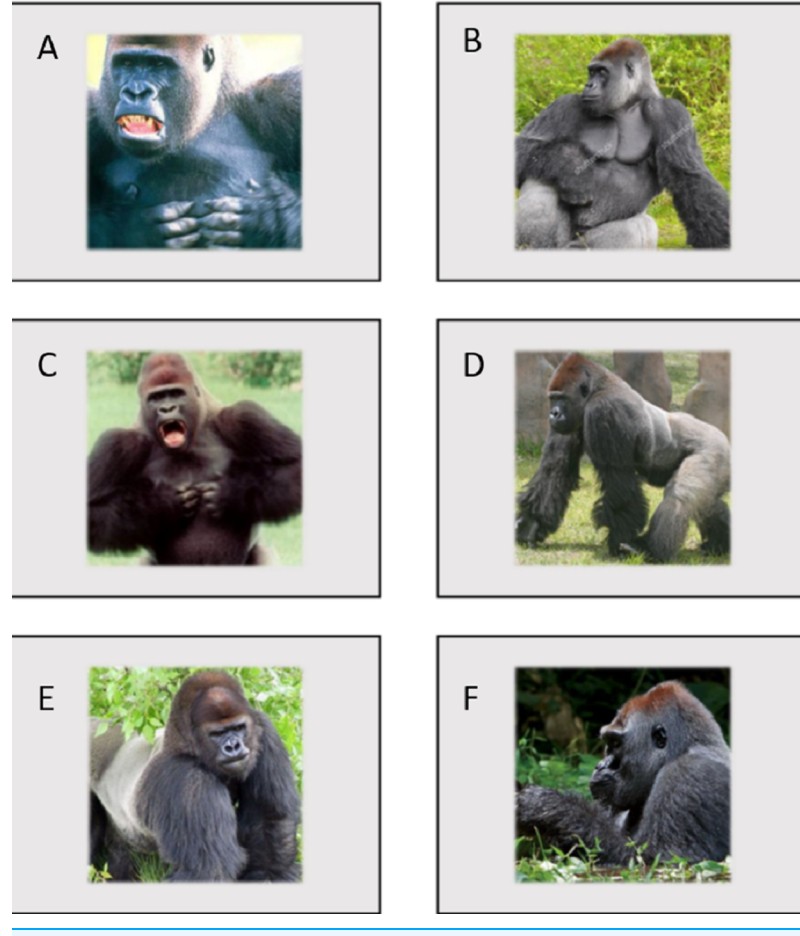

**Figure 1 Images used in Experiment 1.** (A, C and E) are threatening images while (B, D and F) are non-threatening images.

considerations such as temperature and weather conditions (see Table 1). If the temperature was above 4.4 °C without heavy precipitation or wind, and barring other maintenance considerations, the gorillas were housed in an outdoor habitat measuring 6,131 m² or 2,043 m². If the temperature was below 1.7 °C, the gorillas were housed in an indoor habitat measuring approximately 205 m² with a height of 6.7 m high. They typically had access to both habitats if temperatures were between 1.7 and 4.4 °C. All areas were equally familiar to the gorillas. Gorillas were housed in the dayroom overnight for three nights and then in two adjoining stalls for one night on a four-night cycle. If the gorillas had spent the night in the dayroom or accessed, they were tested in single adjacent stalls. Stalls measured approximately 2.1 m wide by 3 m deep by 2.3 m high. If they had spent the night in individual stalls, then they were given access to two stalls and thus, they were tested with more space between them. Thus, proximity to conspecifics during testing covaried with nighttime condition.

Test sessions consisted of 50 trials and took about ten minutes to complete. These trials consisted of nine threatening stimulus trials, nine non-threatening stimulus trials and 32 baseline trials (see Fig. 3). The nine threatening stimulus trials consisted of the three

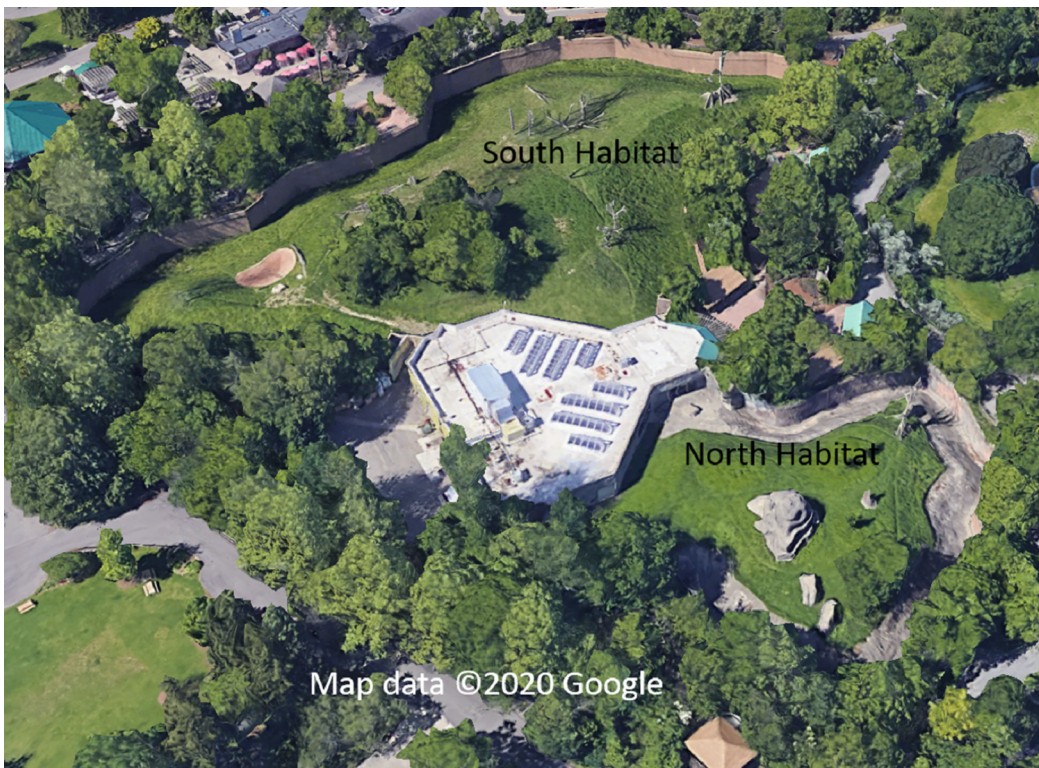

**Figure 2  An overview of the outdoor gorilla habitats from Google Earth.** The South habitat is on the top/left side of the building and the North habitat is on the bottom/right side of the building, which contains the indoor habitats and testing area (Map Credit: © 2020 Google).

**Table 1  Number of sessions for each subject in each condition across Experiment 1.**

| Nighttime | Daytime | | |
|---|---|---|---|
| | Indoors | Outdoors | Accessed |
| Stall | 5 | 12 | 4 |
| Dayroom | 4 | 6 | 4 |
| Accessed | 1 | 10 | 2 |

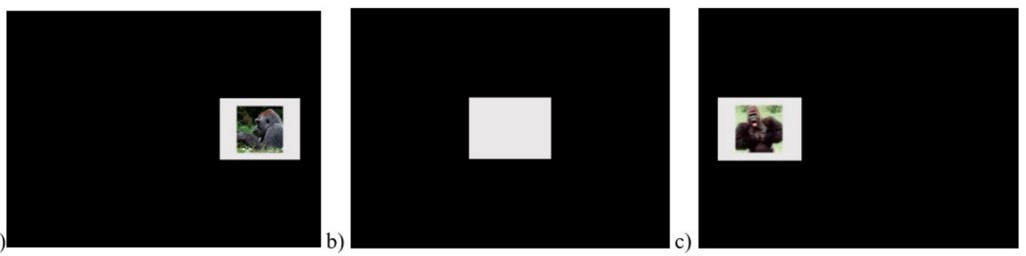

**Figure 3  Example test trials.** (A) Non-threatening, right, (B) baseline, center and (C) threat, left stimuli.

threatening stimuli presented once in each of three locations on the screen (left, center, right). The nine non-threatening stimulus trials consisted of the three non-threatening stimuli presented once in each of three locations on the screen (left, center, right). The 32 baseline trials consisted of the grey square presented in each location 11 times (with one missing trial due to programming error). The first three trials of every session consisted of baseline stimuli presented once at each of the three locations. After the first three trials, the remaining trials were presented in random order with each type of image appearing equally often in each of the three locations on the screen. As in Training, the gorillas were presented with a piece of chow or small piece of produce randomly for touching the displayed stimulus.

## Results

### Analyses

Any outliers (trials in which the latency to touch the screen exceeded two standard deviations (SD) above the mean for that type of trial) were removed. Any trials in which the response was faster than 60 ms were also removed, because this was judged to be the minimum time during which a subject could view the stimulus. A histogram of response latency revealed a positive skew, which was confirmed by skewness statistics reported by SPSS that ranged from 2.16 to 3.38. Kolmogorov–Smirnov tests also revealed that latency data for all subjects violated assumptions of normality with $p$'s all <0.001. Thus, latency data was transformed using a $\log_{10}$ transformation. Histograms revealed improved normality following the transformation. A linear mixed effects model was conducted in SPSS 24.0 with subject as a random effect and stimulus type (non-threatening, baseline, and threatening), daytime location (indoor, outdoor, accessed), and overnight location (stall, dayroom and accessed) as fixed effects. We also included all two-way interactions between stimulus type, daytime location and overnight location as well as the three-way interaction.

### Effects

There was a significant effect of stimulus type ($F_{2,\ 6265.01} = 3.603$, $p = 0.027$), daytime location ($F_{2,\ 6265.05} = 5.773$, $p = 0.003$) and nighttime location ($F_{2,\ 6265.09} = 54.663$, $p < 0.001$) on latency. However, estimates of fixed effects revealed that the gorillas did not touch the threatening stimuli ($M = 3.060$, SEM = 0.037) more slowly than the baseline stimuli ($M = 3.089$, SEM = 0.036, $t = 0.79$, $p = 0.43$) or the non-threatening stimuli ($M = 3.069$, SEM =0.037, $t = 0.92$, $p = 0.93$). The main effects of daytime and nighttime location were qualified by their significant interaction ($F_{2,\ 6265.04} = 21.152$, $p < 0.001$), which is depicted in Fig. 4. If the gorillas were indoors during the day, there was a significant effect of nighttime location in that latencies to respond after spending the night in their stalls ($M = 3.015$, SEM = 0.040, $t = -2.308$, $p = 0.021$) and dayroom ($M = 2.944$, SEM = 0.040, $t = -4.113$, $p < 0.001$) were faster than latencies when they had access to both indoor and outdoor areas ($M = 3.209$, SEM = 0.048). Similarly, if the gorillas had access to both indoors and outdoors during the day, latencies to respond after spending the night in their stalls ($M = 3.036$, SEM = 0.034, $t = -2.721$, $p = 0.007$) and dayroom

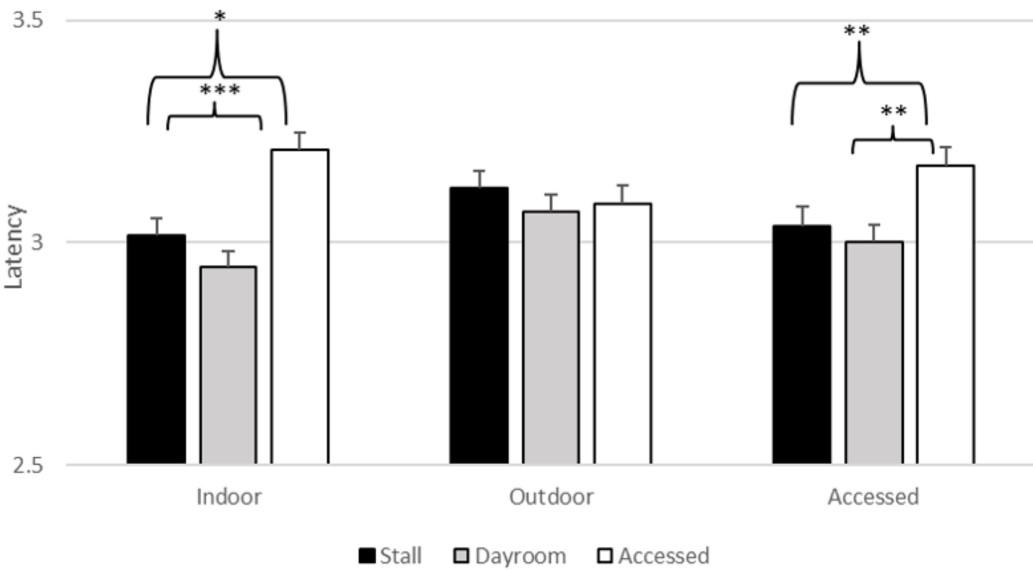

**Figure 4 Average log transformed latency to touch stimulus (across all stimulus types) as a function of daytime and nighttime conditions in Experiment 1.** $*p = 0.05$, $**p = 0.01$, $***p < 0.001$.

($M = 3.001$, SEM = 0.004, $t = -4.113$, $p < 0.004$) were faster than latencies when they had access to both indoor and outdoor areas ($M = 3.173$, SEM = 0.038). If the gorillas had been outside during the day, then there was no significant difference in latencies between the accessed condition and the other two conditions. The interactions between stimulus type and locations were not significant. Thus, the effects of daytime and nighttime housing conditions were not unique to the threatening stimuli[1].

## Discussion

Although there was an overall significant effect of stimulus type, it was not in the expected direction of responding more slowly to the threatening stimuli compared to non-threatening and control stimuli. Furthermore, although the gorillas responded with different latencies to the stimuli depending on where they spent the previous day and night, these conditions did not interact with stimulus type. That is, they did not show response slowing to the threatening stimuli in particular. It is reasonable to question whether the lack of effect of housing condition may be due to the fact that housing conditions do not have long lasting effects. For example, one might expect stronger effects of currently experienced conditions, compared to conditions that were experienced in the hours (or day) preceding testing. However, the lack of response slowing is consistent with that of *Cronin et al. (2018)* who did not find significant response slowing for gorillas or chimpanzees during a presumably aversive air show that they were experiencing at the time of testing. In addition, nighttime conditions did impact current housing conditions during testing as the gorillas had additional space between them during testing if they had been housed in the stalls overnight.

The pattern of data observed here is also consistent with findings from *Bethell et al. (2019)* in which macaques showed behavioral inhibition (failure to respond at all to some

[1] Individual GLMs were also run and the same pattern of daytime by nighttime interactions was observed for each gorilla.

stimuli) but not response slowing, which might cause one to call the procedure into question. However, it is possible that the lack of effect was due not to a failure of the procedure itself, but of the stimuli we selected. Perhaps the stimuli intended to be threatening or non-threatening were not perceived as such by the gorillas. Furthermore, it is possible that we did not obtain the expected effects of response slowing to threatening stimuli because of the large number of sessions, which may have led to habituation to the threatening stimuli. However, in order to determine whether we might find effects of stimulus type in the early sessions, we ran the models for the first eight sessions alone. We could not include conditions as predictors given the lack of balance of conditions. There was still no main effect of stimulus type.

Although we did not obtain significant response slowing to the threatening stimuli, we did observe an interaction of day and night conditions on latencies to respond. The gorillas responded more slowly across stimuli when they had access to both indoor and outdoor areas overnight compared to when they were locked in their stalls or the dayroom, but only if they had been locked indoors or had access to both spaces overnight. When they had spent the day locked outdoors, there was no effect of nighttime location. Interestingly, this hints at slower responses when coming inside from access to a larger amount of space, complexity and choice of habitat, which are generally viewed as positive conditions (*Kurtycz, Wagner & Ross, 2014*). As it is less clear what general response slowing to all stimuli would mean for determining affect, and given the small sample, it might be misleading to draw conclusions based on these findings.

Originally, we had intended to contrast north and south habitats within each of the other conditions as well, thus explaining the large number of sessions and the fact that we could not perfectly balance the nine different conditions within nine-session blocks. The location of the gorillas was not under our control as it was subject to husbandry requirements, weather and temperature. Although access to the outdoor habitats was limited to days when the temperature was above 4.4 °C such that there were no "outside" or "accessed" daytime conditions in the winter, there were such sessions from early spring to mid fall. There were also warm weather days when the gorillas were locked indoors due to other maintenance requirements, so it is not the case that the conditions were completely confounded with weather, although there was certainly a very close relationship between weather and housing conditions. Thus, it is possible that any condition-related effects were attributable to seasonal or weather effects.

Perhaps a more significant limitation is that we did not have access to data regarding the gorillas' behavioral interactions during the time of testing. It is highly likely that agonistic encounters occurring in the time prior to testing had a larger influence on response latencies than the housing conditions did, thus masking effects of our conditions. However, agonistic encounters should not have reduced the likelihood of detecting response slowing to the threatening stimuli in general.

In an attempt to address the issue of salience and habituation to the photos used in Experiment One, we created a novel set of stimuli (Fig. 5) and conducted a reduced number of sessions (8) with these new stimuli in Experiment Two.

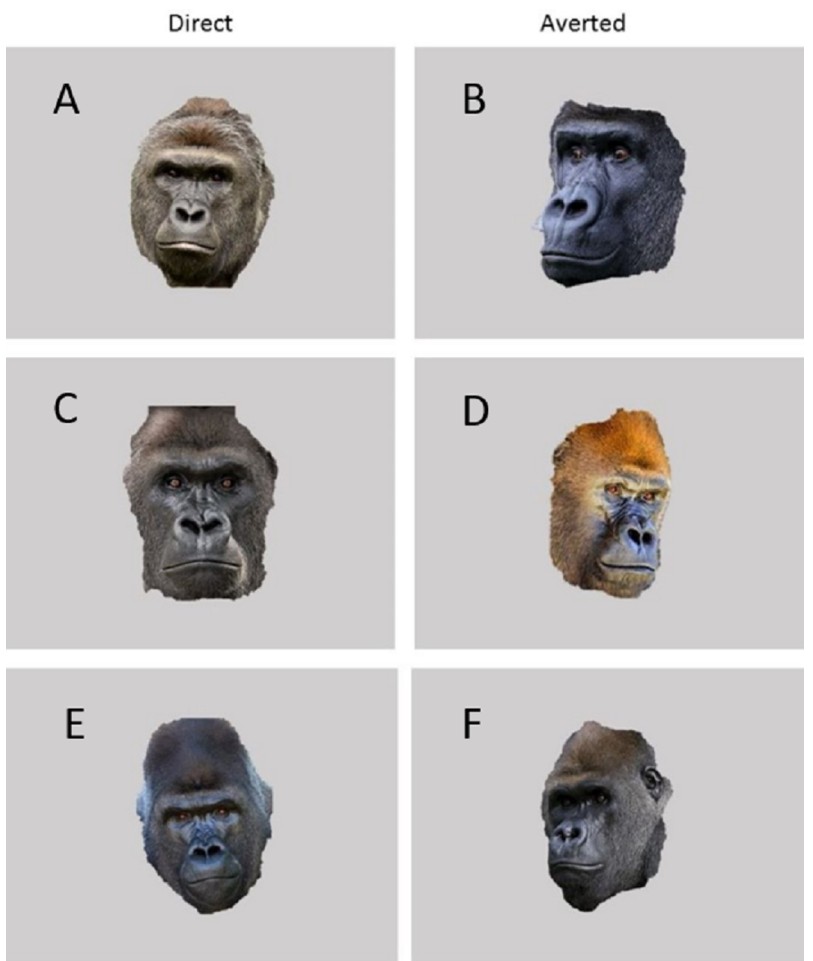

**Figure 5 Images used in Experiment 2.** (A, C and E) are direct/threatening images while (B, D and F) are averted/non-threatening images.

## EXPERIMENT TWO

### Method

The same gorillas participated in Experiment Two using the same apparatus and procedure. The only difference was that the threatening and non-threatening stimuli were replaced with novel images also gathered from non-copyrighted images available via the Internet (see Fig. 5). In this experiment, we operationalized threatening and non-threatening stimuli more conservatively as direct versus averted gaze (as in *Bethell et al. (2016)*) and used images of only gorilla faces rather than their entire bodies. Moreover, the conditions differed slightly because testing took place from December 2017 to April 2018 so there was no outdoor only condition during the day and no accessed condition overnight. Only eight sessions were conducted to reduce habituation to the photographs. Two sessions were conducted within each of the combination of conditions (daytime; dayroom, access; nighttime; dayroom, stall) in random order based on husbandry constraints. All gorillas participated in all sessions.

## Results

As in Experiment One, latencies more than two SDs above the mean for that stimulus type were removed from the data set. The latency data was again subjected to a $\log_{10}$ transformation due to the positive skew. The same linear mixed effects model used in Experiment One was conducted on latency data from Experiment Two. In this model, there was only a significant effect of stimulus type ($F_{2, 1177} = 3.467$, $p = 0.032$). The gorillas responded more slowly to threatening stimuli ($M = 3.267$, SEM = 0.074) compared to baseline ($M = 3.211$, SEM = 0.071) and non-threatening stimuli ($M = 3.174$, SEM = 0.074), but estimates of fixed effects indicated that these differences were not significant. There were no effects of daytime or nighttime location and no interactions.

## Discussion

In this experiment with novel stimuli and fewer sessions, we did observe the expected main effect of stimulus type. The pattern of results revealed response slowing to the threatening stimuli. Thus, it seems important to reduce repeated exposure to the stimuli in order to obtain a response slowing effect. Future studies should ideally introduce novel stimuli in each session. Of course, the results may also indicate a better selection of stimuli in Experiment Two compared to Experiment One. Here, we focused only on whether gorillas were facing forward or had gaze averted rather than making assumptions from posture etc. as to how the images would be perceived by the gorillas.

Although these results encourage the use of the response slowing paradigm to some extent, response slowing was not influenced by the housing conditions experienced by the gorillas. This experiment was conducted over a shorter period of time (4 months) without the daytime outdoor only and accessed overnight conditions, and it is possible that there was less variability in gorilla affect during this time. Again, we did not have access to observations of their behavior or use of space during this time.

## GENERAL DISCUSSION

Response slowing has been used as a measure of negative affect (fear or anxiety) in both human and nonhuman subjects. For instance, people that displayed Looming Cognitive Style (a tendency to interpret stimuli as threatening or dangerous) also displayed increased reaction times when presented with images of animals, even when those animals were nonthreatening or appeared to be moving away from the viewer (*Riskind et al., 2016*). Similar results have been found in nonhuman primate species. For example, *Bethell et al. (2016)* found that after undergoing veterinary care (assumed to result in negative affect), rhesus macaques displayed a similar trend of response slowing, in this case increasing their reaction times to threatening images of conspecifics making direct eye contact. *Cronin et al. (2018)* also found that Japanese macaques showed greater response slowing during a loud air show compared to control trials. However, chimpanzees and gorillas tested under similar circumstances showed more subtle effects. As in Experiment One here, it is likely that the apes habituate quickly to the stimuli, indicating the need to present constantly changing stimuli in these designs. Stimuli must also be selected carefully, making dimensions that are important to the study species salient.

Somewhat inconsistent support for the response slowing model may be due to the fact that individuals vary in their sensitivity to threat. *Bethell et al. (2019)* found individual differences in response slowing. In their first study, individuals that showed response slowing were those that displayed more significant amounts of freezing in the HIT, verifying that the response slowing may indicate individuals with higher levels of dysregulated fear response. Previous studies have found that dominant animals exhibit higher levels of optimism in judgement bias tasks (tufted capuchin monkeys, *Schino et al., 2016*; rats, *Barker et al., 2017*). *Schino et al. (2016)* did not find short-term effects of increased optimism after receiving bouts of grooming; however, individuals that overall received more grooming were more optimistic. An understanding of the interaction of social status and response to space could inform management of captive species by helping to predict an animal's behavior and preferences based on their dominance status. Here, we tested only three gorillas so we are limited in the extent to which we can generalize our findings to other gorillas or other housing conditions. Future studies would benefit from an inclusion of behavioral and perhaps hormonal data along with cognitive testing.

We had assumed that the gorillas would feel less stressed and demonstrate less response slowing when provided access to a larger, more complex amount of space, and when given a choice as to which space to occupy. Somewhat surprisingly, we found that gorillas responded more slowly under conditions where they had more space. This result is not altogether unexpected, however, *Cordoni & Palagi (2007)* found that gorillas did not demonstrate greater conflict in more dense spaces. In fact, they argued that the gorillas adapted their behavior to the increased density by increasing positive behaviors such as touching and reconciliation. These arguments have also been made in studies with rhesus macaques and chimpanzees (*De Waal, 1989*; *De Waal, Aureli & Judge, 2000*; *Duncan et al., 2013*; *Nieuwenhuijsen & De Waal, 1982*). Other studies have also shown that gorillas may be less likely to fully utilize larger habitats compared to even chimpanzees (*Ross et al., 2011*). Thus, it is possible that they prefer greater proximity and smaller spaces where they can more closely monitor potential threats.

There may be other reasons for the complicated relationship between space and aggression in primates (*De Waal, 1989*; *Hosey, 2005*). *Alexander & Roth (1971)* suggested an inverse relationship of environmental space and aggression in Japanese macaques. They found that the macaques displayed increased aggression in response to a reduction in available space, but theorized that this aggression, at least in part, may have also been due to the unfamiliarity of the new space. Similarly, *Southwick (1967)* demonstrated an inverse relationship between space and aggression in rhesus macaques, yet he also suggested that another factor, social changes, may have had a stronger effect than space alone. In the current study, familiarity and social structure were not confounds; all of the environmental spaces, regardless of size, were familiar to the gorillas and there were no changes to the social group. The fact that they experienced all conditions frequently on alternating days may help to explain the lack of a consistent effect of housing conditions. In addition, access to outdoor space was confounded with seasonal and temperature changes, which may also correlate with hormonal and dietary changes, such as reduced

access to forage during winter months. *Fuller et al. (2018)* found beneficial effects of increased foraging materials in the same group of gorillas tested here. Thus, apparent effects of space may be due to other unconsidered variables instead.

It should be noted that although our conditions could generally be taken to represent different amounts of space, the habitats could not solely be reduced to this factor. Outdoor areas also provided different types of enrichment compared to indoor spaces. Different areas vary in their complexity, which may be a predictor of aggression (*Hoff et al., 1997*). Being housed in stalls versus the dayroom overnight also differed in opportunity to interact and the need to be vigilant to threats from conspecifics. Accessed conditions provided not only more space, but also more choice, which has also been shown to be important to animal welfare (*Kurtycz, Wagner & Ross, 2014*; *Ross, 2006*). Thus, we believe it would be simplistic to interpret the findings simply in relation to available space. Ideally, a study could control all of the potentially important elements, but that is not typically the case when conducting research outside of the lab. Although the current study has limitations, we believe it contributes to the validation of a promising new assessment tool.

Although not immune to interpretation ambiguities, the response slowing method of measuring affect may be the best method available for subjects that are ill-suited to the extensive training required for other methods (such as the methods described in previous experiments; *McGuire & Vonk, 2018*; *McGuire et al., 2017*; *McGuire, Vonk & Johnson-Ulrich, 2017*). Because there is minimal training necessary to implement this measure, it could be a quick and efficient tool for measuring animal affect. With very little training, it was possible to investigate the potential for changes in cognitive bias, as evidenced by response slowing across various seasonal changes in habitat and husbandry routines.

It is also possible that, as this measure makes use of an automatic attentional process, that it could be used to identify individuals that may be at risk of developing more problematic symptoms of negative affect, such as stereotypic behaviors, before they develop (*Bethell et al., 2016*). A similar measure could be used to identify problem animals and intervene through positive changes in the management and husbandry of the animals before a true problem arises. This would be particularly useful for species that are prone to stereotypic behaviors that arise due to stress or poor welfare conditions, which can be notoriously difficult to extinguish once expressed (*Mason, 1991*).

## CONCLUSIONS

The response slowing paradigm will need to be more thoroughly validated before it can be reliably used as a tool for animal welfare researchers through extensive testing and replication, but it is one of the more promising new methods available. One question will be how this paradigm can be adapted to suit species for which computer-based assessments are less feasible, such as equids. One possibility would be to utilize approach times to aversive and preferred feeding locations. Studies have shown that an animal's foraging pattern may be related to their perceptions of safety (*Shrader et al., 2008*; *Troxell-Smith et al., 2017*; *Troxell-Smith et al., 2017*). The validity of this measure could be

determined by looking for other indicators, such as hormonal or behavioral indicators. For instance, the validity of this measure would be supported if hormonal assays found that hormones related to stress, such as cortisol, or instances of abnormal behaviors, such as hair plucking, were also reduced in response to the gorillas spending time in smaller spaces. In sum, this is a promising method that awaits further validation in a larger number of subjects and species. The jury is still out on its utility for testing apes.

## ACKNOWLEDGEMENTS

We wish to thank the Great Ape keepers and Center for Zoo Animal Welfare at the Detroit Zoo for supporting this research.

### Funding
The authors received no funding for this work.

### Competing Interests
Jennifer M. Vonk is an Academic Editor for PeerJ.

### Author Contributions
- Molly McGuire conceived and designed the experiments, performed the experiments, prepared figures and/or tables, authored or reviewed drafts of the paper, and approved the final draft.
- Jennifer M. Vonk conceived and designed the experiments, performed the experiments, analyzed the data, authored or reviewed drafts of the paper, and approved the final draft.

### Animal Ethics
The following information was supplied relating to ethical approvals (i.e., approving body and any reference numbers):

The Institutional Animal Care and Use Committee of Oakland University approved this research (12063-R1-A1).

### Data Availability
The raw data and a codebook for the raw data are available in the Supplemental Files.

### Supplemental Information
Supplemental information for this article can be found online at http://dx.doi.org/10.7717/peerj.9525#supplemental-information.

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
