# Peer review of "In or out: Response slowing across housing conditions as a measure of affect in three Western lowland gorillas (Gorilla gorilla gorilla)"

_PeerJ, doi:10.7717/peerj.9525_

## Round 0.1 · original submission · Major Revisions

Thank you very much for your submission to PeerJ. I have been fortunate to receive three reviews from experts in your field. All three reviewers have provided thoughtful commentary on your article and I encourage you to carefully consider each of their suggestions as you revise your article. While all three reviewers praised the clarity of your writing and thoroughness of your background literature presentation all three also had some major concerns with your article that must be addressed before I can consider this article for publication.

I will not reiterate the reviewers' comments here, but I did want to highlight a couple of common themes across the reviews. In particular the reviewers noted that the conclusions you can draw from these two experiments are limited by the small sample size you tested (3 gorillas) and a number of confounding factors. While I am sympathetic to the nuanced nature of conducting research in a zoo setting and the opportunistic nature of your research paradigm, I do however request that you more fully acknowledge these limitations in your article and the fact that they reduce the broader level conclusions you can draw, as well as the interpretations you make about individual differences. Additionally, the reviewers asked that you clarify your methods and terminology throughout and I agree that this would help.

In addition to the reviewers' comments, I had a few items that I ask you to address too:

Given that your key experimental variable for both experiments was the different housing options that the gorillas experienced you need to describe each of these in much more detail as well as the gorillas' experience with them (i.e. how often they get to go outside or have access to the different spaces). Understanding the qualitative and quantitative nature of these spaces is important as well as their "novelty" for the gorillas.

For experiment 1, I am not sure why you separated out the housing combinations into a 3x2 design (3 daytime locations x 2 nighttime locations). It seems more parsimonious and biologically meaningful to combine the day and nighttime locations, such that you have 6 experimental conditions, such that this encapsulates the entire 24 hours prior to testing.

For both experiments it was not clear to me whether the number of test sessions you ran per condition was equal for all subjects tested. If not, can you please provide the average of sessions per subject per condition (+SD) rather than the total number of test sessions per condition. (Related to this, please confirm that all subjects were tested in all conditions in both experiment.)

Lastly, for balance, please provide the stimuli that you used in experiment 2 as well as for experiment 1 (I could not find them as supplemental materials). An in-text figure as you have for experiment 1 would also help here.

I look forward to receiving your revised article and your responses to the reviewers' feedback.

Reviewer 1 ·

Basic reporting

This is an interesting and clearly written study about gorilla affective states. The background literature is well understood and covered.

The word "states" is missing from line 1 of the Abstract.

"these individuals spend a relatively high proportion of time
vying for dominance and it may be that they experience a tradeoff concerning these two sleep locations"
please provide data to support this suggestion.

Experimental design

The experimental design appears robust and is clearly described.

My first main concern is the sample size of 3. I appreciate that great ape studies are typically limited in terms of the possible sample sizes available, but given that the study overall finds no consistent effects, it may just be a sample size issue. This has not been discussed.

Validity of the findings

My other main overall concerns is that the effects of one previous night on the current mood of the gorillas might not be strong enough to detect an effect. It seems that the night time conditions/enclosures for the three gorillas were changing quite often. Details of how often changes occurred and where the gorillas spent their nights (husbandry routine of the zoo) should be given early in the Methods. There is some information currently between lines 165 and 171. I could not find Table 1 among all the submitted files (see line 171).

Given that the gorillas seem to have been moved often between preferred and less-preferred night enclosures, this might explain the lack of consistent effect. This possibility has not been discussed.

Additional comments

n/a

Reviewer 2 ·

Basic reporting

For the most part, the article is written clearly. However, the abstract needs revision (see comments below). In addition, the article seems to be missing a large portion of background literature relevant to their hypothesis regarding the effect of housing condition on affect. See specific comments below.
The article is structured appropriately with the exception of some methodological details that should be moved from the introduction to the method section, and there are several points that need to be expanded upon in the method section (see general comments below). Additionally, the structure of the discussion should be revised to improve readability and the flow of results in the context of previous literature and alternative hypotheses. As it stands now, the discussion jumps around quite a bit between different findings, alternative hypotheses, and background literature. See comments below.

Experimental design

To me, it seems that there are two knowledge gaps that this study is attempting to address: 1) testing the validity of the response slowing paradigm in measuring ape affect, 2) whether housing conditions impact gorilla affect, as measured through the response slowing paradigm. I think this needs to be made clearer in the introduction. More importantly, the authors must include more background information regarding their second aim (the effect of housing condition on affect). Lastly, and perhaps most importantly, the hypothesis that gorilla affect will be affected by housing conditions seems to be largely confounded by weather / season, as is even mentioned by the authors. However, they do not attempt to address this. I have expanded on each of these issues below.

Validity of the findings

My biggest concerns are related to the validity of the findings. Because of the apparent confounds mentioned above, I am doubtful of the ability of the authors to make any solid conclusions about their hypotheses and about the validity of this experimental paradigm. The purpose of this study was to use the response slowing paradigm (which is dependent on longer latencies to respond to threatening stimuli) to assess the effects of available space. However, not only did the results fail to show longer latencies to response to threatening stimuli as a function of housing condition, the gorillas didn’t show longer latencies in response to threatening stimuli in general. If the paradigm is dependent on response slowing to threatening stimuli, but gorillas didn’t do this, how can conclusions be made regarding the effect of space and housing? More generally, what does this mean for the idea of “response slowing” as a measure of affect? What does this mean for hypotheses regarding available space? While the authors give several alternative explanations for their results (e.g., increased attention, dominance, vigilance, habituation to stimuli), they do not explicitly address the above issue.
A paragraph detailing the shortcomings of this study, along with ways that future research could address these issues, is needed in the general discussion, particularly regarding 1) the low N, 2) the housing condition confounds of temperature, husbandry procedures, and season. Additionally, there seem to be many alternative explanations that need to be highlighted in the discussion. Weather, environmental complexity, individual space preferences, social proximity, husbandry routines, the lack of investigation of interactions between daytime and nighttime housing, and coping mechanisms during prolonged periods of space restriction (see sources listed above) should be discussed in the context of previous research.

Additional comments

Line 21: please correct “affective” to “affect”.
Line 21-22: “response” appears twice, please revise to make less repetitive.
Overall, the abstract needs some work to improve detail and clarity. Specifically:
-The background portion of the abstract needs more detail. What is response slowing? Does “individuals” refer to humans, primates, other animals?
-Please provide more detail regarding how the housing arrangements at the time of testing were impacted by daytime and nighttime housing arrangements. This point is also difficult to follow in the body of the manuscript (Lines 165-173).
-Please specify N for the experiment.
-Please clarify whether the first sentence in the “Method” portion of the abstract refers to the general method for both experiments.
-More detail is needed regarding “three daytime and nighttime conditions” in line 26-27, and “four conditions composed of daytime and nighttime arrangements” in line 28. These are supposed to correspond to varying levels of space restriction, but this is unclear here.
-In my opinion, the claim cannot be made that the “results are somewhat consistent with those of a previous study” (Line 31-32) given that just one subject showed response slowing across two experiments.
Line 41-42 “…most elusive of the states…” please clarify what other states to which this refers.
In line 46, the authors state the judgement bias paradigms may not assess affective states at all, but then go on to say in Line 53-54 that “Previous studies have shown that judgement bias tests can be sensitive to affect changes….” Please clarify the difference here.
Line 59-60. Please expand on the theory or background behind response slowing. What does it aim to test? What does a slowing in response indicate, or why is response slowing indicative of negative affect? How does that relate to the “behavioral inhibition” that is mentioned later in the paragraph?
Line 55-58. Please remove current study procedure from this background literature paragraph. It feels like the introduction bounces around between current study aims/methods and background information (including line 101-109 as well). Therefore, it may be helpful to move all current study information to the last paragraph of the introduction and/or the method section. Specifically, line 104-109 seems that it would be more appropriate in the method section.
Line 110-113. Much more background literature regarding the effects of housing on behavior and welfare must be included here. Differences in housing are at the core of their hypothesis, but there is no background information to support how they came to that hypothesis. For example, the article is missing some critical background information regarding apes’ coping strategies in response to confined spaces. Although much of this background is in chimpanzees, I believe it is highly relevant here and I have included some sources below. The predictions regarding space also make no mention of space complexity, which, some have argued, is more important than amount of space alone. I would like to see additional detail regarding these points in the introduction, which may likely inform their hypotheses as well.
Importantly, the author’s predictions are largely confounded by differences in weather in their “housing conditions”. As they state in line 115 – 119, the hypothesized increase in positive affect could occur due to either 1) access to larger spaces, or 2) warmer weather, 3) a combination of these. The authors make no attempt at discussing this confound and how it might impact their data.
• Aureli, F., & de Waal, F. B. M. (1997). Inhibition of social behavior in chimpanzees under high-density conditions.
• de Waal, F. B. (1989). The myth of a simple relation between space and aggression in captive primates.
• de Waal, F. B., Aureli, F., & Judge, P. G. (2000). Coping with crowding.
• Duncan, L. M., Jones, M. A., van Lierop, M., & Pillay, N. (2013). Chimpanzees use multiple strategies to limit aggression and stress during spatial density changes.
• Herrelko, E. S., Buchanan-Smith, H. M., & Vick, S.-J. (2015). Perception of available space during chimpanzee introductions: Number of accessible areas is more important than enclosure size.
• Nieuwenhuijsen, K., & de Waal, F. (1982). Effects of spatial crowding on social behavior in a chimpanzee colony.

Line 131: Dominance struggles are mentioned several times throughout the paper. Please include dominance status of each individual if the data are available.
Line 133-135 should be included in the procedure section. I would like to see additional housing information regarding the types and amounts of spaces available to gorillas in the method section. Were these holding areas in which gorillas were tested the individual stalls where they sometimes slept? Also, what is the size of these testing areas? It would immensely helpful to include a diagram of their housing, including all areas mentioned in the method section.
Line 165-173: This section should be revised and more detail added to clarify how the housing combinations created the “conditions” that the authors tested. As mentioned above, please include additional information from the introduction (lines 109) on the amount and type of space available in each of these housing conditions in the method section. Please provide additional detail regarding the husbandry considerations, weather, and temperature that determined these housing conditions. What affected nighttime locations? Specifically, why would gorillas sometimes be locked into the stalls instead of the dayroom? Lastly, do the authors have any information regarding social proximity in these housing conditions, specifically at night? For example, did the gorillas tend to sleep farther apart in the “accessed” condition?
What time of day did testing occur?
What are the dates of testing for experiment 1?
Approximately how long did it take the gorillas to complete their 50 trials for each session?
Line 186-187: how often did that occur?
Line 194-197: Why are there no expectations and analyses regarding interactions between the daytime and nighttime conditions? This seems like a logical next step that being locked indoors for the day and then sleeping in a stall overnight might result in the slowest response times (given a longer period of access to smaller spaces, according to their predictions). Additionally, because they did not examine interactions, they cannot rule out the possibility that previous daytime housing conditions drove the effect of what they attributed to previous nighttime housing condition (or vice versa). For example, because they collapsed across nighttime conditions to examine the effect of daytime conditions (that is, looking only at the main effect of daytime condition), perhaps the effect of slowest response time in the daytime “accessed” condition was driven solely by the nighttime “stall” condition. At the least, a discussion of these possibilities is warranted.
Line 233-243: It may be helpful to organize the discussion such that group-level effects are discussed first, followed by individual-level effects. So this paragraph should perhaps focus only on the group-level effects. Additionally, the discussion should attempt to focus on one idea at a time. For example, the first paragraph introduces increased vigilance, previous study findings, and talks about both housing condition and stimulus type effects, all of which are brought up again later in the discussion. In my opinion, each paragraph should detail a specific effect and a discussion of the relevant background info/previous studies, ideas about why the effect occurred, etc. As the discussion is currently, it bounces around between group and individual effects, and introduces new concepts without detailing them (e.g., line 253-255 talks about spacing during testing, but does not expand on why this might affect results).
Additionally, the discussion could be strengthened by including information from some of the space and complexity studies noted above. I think some of the findings may be explained in terms of findings from these previous studies.
Line 241: Please rephrase this sentence such that it is not a double-negative, and expand on what the consistency between the results of the current study and Cronin et al. may mean for the test paradigm, species, etc.
Line 246: by “in general” do you mean “descriptively”? Because not all of the gorillas showed a significant difference?
Line 298: how are the conditions random if they were also based on husbandry constraints?
Line 305: typo “M” before the parentheses?
Line 326: couldn’t this be another explanatory variable for the housing condition findings in experiment 1? Perhaps the previous day’s agonistic encounters affected response time?
Discussion:
Line 347: Could another alternative explanation be the choice of stimuli used in the studies? Could the threatening stimuli used perhaps be not as aversive as the author’s assumed? And perhaps the non-threatening stimuli had some sort of emotional valence attached, whether positive or negative?
Line 348: “It is also possible that apes are generally less sensitive to environmental changes compared to monkeys.” I don’t think the literature supports this (see citations above as well). I think a larger discussion regarding space availability, complexity, and coping mechanisms in response to confined space is needed here.
Line 355-357: This same argument has been made by de Waal and colleagues (see citations above). Again, I think these studies would help to explain the results found in the current study, and this should be elaborated upon in this discussion.
Line 371-374. This is a huge confound in the experimental design and results, in my opinion. This point should be elaborated upon: how might this have affected the results of both experiments?
Line 390-405: These paragraphs should be stated much more tentatively. Given the confounds with housing condition in the current study, low N, some null findings, subject-dependent effects, and findings that were in opposition to the hypotheses and previous research, a larger discussion needs to be included about the validity, efficacy, and utility of this paradigm in assessing ape affect (including the points I just mentioned).
The finding that each gorilla showed a different pattern of results should be discussed in the context of the validity of the paradigm. Does the fact that they were all different tell you anything about individual housing preferences? Does it tell us anything about the validity of this paradigm?
Figures:
Fig 2: Mismatch between legend and bar colors.
For all figures, please include brackets corresponding to asterisks showing significance. For example, in figure 4, which conditions are different from one another and correspond to each asterisk set?
Fig 6: Please outline the “access” condition, or use a different color as the bar is difficult to see. Additionally, how is Pende’s latency different between dayroom and access if the error bars overlap significantly? Is this typo? If not, this should be addressed in the discussion or legend.

·

Basic reporting

The manuscript is well-written, and the introduction and discussion integrate relevant literature. The potential strengths and weaknesses of the paradigm are clearly explained and the data contributed by the current study are helpful to the field. I have no concerns regarding the article structure.

Experimental design

One weakness of the experimental design (or the presentation of the experimental design) is that it is difficult to follow how the housing conditions differ from one another along the key dimension of available space. This confusion starts in the abstract, where “nighttime” and “daytime” housing are introduced as the variables of interest, rather than available space which isn’t mentioned until later. The authors do a great job of explaining how and why bachelor gorilla affect may be impacted by amount of available space, but the conditions “indoors/outdoors/accessed; daytime and nighttime” are difficult to track along this meaningful dimension. Minimally, a figure that shows the available spaces and how different scenarios / combinations relate to the cells in Table 1 would be useful (this is raised in comments below as well). Alternatively, classing conditions based on amount of long-term and short-term social space could be more intuitive, and potentially change the model structure (see comment immediately below).

Related, I think the potential impact of timeframe needs to be considered in some way. The daytime access is a long-term factor (seasonal, if I understand). The nighttime access is a recent short term factor (immediately previous night). There is a third way that social space enters in, and that is the immediate presence of a conspecific nearby (or not) while testing. Given the effects observed in previous studies, is there a logical prediction about timeframe that could be incorporated here (? Did the authors consider presence/absence of a neighbor during testing as its own effect in the model? (Unless this covaries 100% with another condition, which I may have not understood, see comment above.) I realize daytime and nighttime are different effects in the model, so statistically they are considered differently, but they are not discussed in this way. Minimally, introducing the different timeframe of these effects of interest would be helpful in the discussion, ideally with reference to previous literature and predictions.

Validity of the findings

I’m not sure the detailed individual subject analyses that follow the group analyses are warranted, or helpful. The authors are questioning the paradigm in the introduction, there are only 3 gorillas, and there aren’t corresponding behavioral data indicating differences that can aid predictions or interpretations. Related, while I appreciate the honesty of the speculative nature of the potential reasons for the individual differences (lines 321-333) they aren’t very compelling as presented. Individual differences way heavily in the discussion (starting line 375), and I think this is still relevant to include. However, it does not need to focus on these specific individuals. Instead the reviewed literature could remain and a call for more study with larger samples and corresponding behavioral or rank-related data.

Additional comments

In this study, the authors applied a response-slowing paradigm to the study of three zoo-housed gorillas from one social group, and questioned whether any response-slowing effects observed were predicted by housing conditions. The authors clearly discuss the past uses of this paradigm and the conflicting results that have accompanied it, and set the stage for this study well. Their results do not overwhelmingly support the use of this paradigm in apes, and leave open two possibilities 1) that the apes were not experiencing negative affect 2) that the apes did experience negative affect but it was not apparent in this test. Despite the complicated nature of the findings, the authors handle the discussion well. In addition to the more substantial comments above regarding the meaningfulness/explanation of conditions and the focus on individual data, I provide the following minor comments as well:

Minor:

Line 75: This is a very clear and important paragraph, but the final sentence seems to focus only on the minimal training and not the limitations explained. Would it make more sense to complete the paragraph with something like “Although there are open questions about the reliability of the response slowing task, the feasibility that comes with minimal training makes it a potentially valuable avenue to continue investigating” (or something similar)?

Line 95-98: this sentence long and a bit hard to follow.

Line 159: technically speaking, the tone was not a reward (not initially at least), suggest saying that it provided auditory feedback to the subject to indicate that the touch was successful and a food reward would be delivered.

Line 281: unclear how the gorillas never failed to select stimuli if one gorilla (Chip) stoped participating during some sessions?

Line 320; The term “dysregulated fear response” is used in the abstract and the introduced in line 320, this should be defined early.

Line 328: using the phrase “as in Experiment One” makes the meaning ambiguous, suggest rephrasing

Line 344: revise “more significant,” perhaps replace with “greater”

Line 352: comma unnecessary

Line 404: it may appear to readers that the “i.e.,” provides a definition of stereotypic behaviors, which it doesn’t. replace with definition of stereotypic behavior, or rephrase

Line 412: some more recent references regarding foraging style as in indicator of welfare may be:

Troxell-Smith, S. M., Watters, J. V., Whelan, C. J., & Brown, J. S. (2017). Zoo foraging ecology: Preference and welfare assessment of two okapi (Okapia johnstoni) at the Brookfield Zoo. Animal Behavior and Cognition, 4(2), 187-199.

Troxel-Smith, S. M., Whelani, C., Magle, S. B., & Brown, S. (2017). Zoo foraging ecology: development and assessment of a welfare tool for. Animal Welfare, 26, 265-275.

Throughout: should the threat stimuli be referred to as “threatening stimuli”? Seems more intuitive and would correspond with “non-threatening stimuli”

---

## Round 0.2 · Minor Revisions

Thank you very much for your resubmission to PeerJ. All three reviewers who reviewed your original submission have kindly provided feedback on this revision.

All three concur that you have responded carefully to all their initial feedback, and I agree that the article is much improved. However, Reviewer 1 raises the valid overarching concern that your study is seriously limited by the small sample size.

I concur with Reviewer 1 that your small sample size limits the conclusions you can draw from your study, although I think the evaluation you provide of these response-slowing methods (in terms of stimuli type and trial number recommendations) will be of value to the field. Therefore, I am recommending acceptance of your article (pending a few minor revisions requested by Reviewers 2 and 3) but I would request that you emphasize this limitation of your study more explicitly. Specifically, I suggest you edit your title to note that it is a case study, or to highlight that you tested three gorillas. Similarly, I think rather than saying "bachelor group" in your abstract you should simply say "three male gorillas". Thus, right from the outset you can present a more upfront presentation of your methods and, therefore, what conclusions can be drawn from your results. In this way, your paper is more clearly a case study on the experiences of these three gorillas at this one zoo, and an evaluation of response-slowing paradigms, rather than a general message on gorillas' affect in relation to exhibit access.

I believe if you can make these adjustments, as well as all the edits recommended by Reviewers 2 and 3, it will be my pleasure to recommend your article for publication. I do not think I will need to send your article back out for review again.

Reviewer 1 ·

Basic reporting

n/a

Experimental design

n/a

Validity of the findings

n/a

Additional comments

Overall, the manuscript has improved.
Am I convinced that this study tells us anything about affect in gorillas? No - because the sample size is still 3, which cannot be changed.

Reviewer 2 ·

Basic reporting

All sections of this manuscript are much improved, and I commend the authors for such careful and well-done revisions. My comments correspond to line numbers from the PDF provided.

The language used, context, background literature, and article structure are well done and appropriate.

Minor suggestions related to Figure 4:
1) Please include in the title that the figure depicts latency across ALL stimuli types.
2) The title includes that “*** = p <.001”, but three asterisks don’t appear in the figure. Therefore, please specify the p-values to which * and ** correspond.
3) Lastly, it is difficult to tell which effect corresponds to each set of asterisks. I think the placement of these, and perhaps additional brackets, would aid in showing which asterisks correspond to which effect.

Line 107-110 now seems out of place. I think the sentence starting on line 109 would be better toward the beginning of the paragraph starting on line 129. The sentence from 107-108 might be better suited later in the last paragraph of the introduction, where the authors talk about threatening stimuli (Line 149). Lastly, editing the sentence starting on line 110 to have more of an emphasis on affect would allow it to serve as a new introductory sentence to the paragraph. Something like…“Previous studies have indicated that different housing arrangements have an effect on ape affect, as measured through behavior.”

119-120: please add a reference for this specific finding.

Experimental design

These areas are much improved. The authors have done a great job in adding methodological detail. There are just a few points regarding methodological detail that remain for me:

In the abstract, “nested conditions” are described, but they are not mentioned or explained in the method section. Please add an explanation of what is meant by nested conditions in the “Testing” section of the Procedure (starting on line 190).

Line 189-194: Please be consistent with terms used to describe housing conditions throughout the manuscript and between the text and figures (e.g., in figure 3, “Indoor” is listed as “Inside”, and “Inside” is used in L230, 242 but the condition is labeled as “Indoor”). Additionally, I understand that perhaps the authors are trying to distinguish between daytime and nighttime conditions, but isn’t the daytime condition “Indoor” the same as the nighttime condition “Dayroom” (i.e., locked in the NDR or SDR)? If so, labeling both of these consistently might aid in reducing confusion between the conditions.

Additionally, in their reply to a reviewer, the authors explain why they chose to analyze the housing conditions separately, rather than creating 9 conditions based on a combination of each daytime and nighttime condition ( “the issue of space when sleeping at night was qualitatively different than the issue of space when foraging throughout the day….”). This makes a lot of sense and explains why they did what they did. As such, I think it would be helpful to include exactly that information, perhaps at the end of the “Testing” paragraph (L189-208).

L218: Perhaps I missed this elsewhere, but can the authors add here that it took approximately 10 min for the gorillas to complete each session? I think this is important given that they were confined to a smaller space during testing.

Validity of the findings

This section is much improved, results are discussed appropriately and in-context, and includes alternative explanations.

Additional comments

In the abstract, I suggest making it clearer that the housing conditions represented varying levels of space and choice restriction. For example, a simple phrase stating that the housing conditions represented different levels of space and choice restriction could be added to the end of this sentence: “In Experiment One, they completed 48 50-trial sessions across combinations of three nested daytime and three nighttime conditions, which represented different levels of space and choice restriction.”

Line 122: Nieuwenhuijsen is misspelled.

Line 194: missing end parenthesis.

Line 300: “condition-related” rather than “condition related”?

Line 371: missing (year) for Schino et al. in-text reference.

Line 383: Please expand on this idea: to which result are you referring? What is meant by tradeoff?

·

Basic reporting

I have no concerns about basic reporting in the revised manuscript.

Experimental design

I have no concerns about experimental design in the revised manuscript.

Validity of the findings

I have no concerns about the validity of the findings or how findings are discussed or interpreted in the revised manuscript.

Additional comments

In this revision, the authors have addressed my earlier concerns. The only comments that I have at this point relate to minor wording corrections/clarifications throughout. Specifically:

Experiment One, subsection Testing: there is no report of what food rewards were used. If they were the same as training (small piece of chow or produce), were these mixed/randomized to avoid inadvertent associations between stimuli type and food? Please add details.

Figure 4: What is the y-axis label?

Line 21-22: Don’t need both parentheses & commas

Line 32: Suggest rephrasing “had been accessed to” to “had access to”

Line 73: Suggest changing “previous” to “recent”

Line 83: Suggest changing “unlike” to “without”

Line 102: Suggest starting sentence with “However, Bethell…” to point out contrast with previous sentence/finding.

Line 342: rather than say these results “validate the use to some extent” suggest saying the results are encouraging, or something similar.

Line 423: rather than “it is a quick and efficient tool,” suggest “it could be a quick and efficient tool” to correspond with cautious support for the approach that has been evidenced to this point.

Are stalls and aisles synonymous?

---

## Round 0.3 · accepted · Accept

Thank you for carefully responding to each of the reviewers' final outstanding suggestions and for also making changes in response to my feedback. It is my pleasure to recommend your article for publication in PeerJ.